# Genetic Animal Models of Cardiovascular Pathologies

**DOI:** 10.3390/biomedicines13071518

**Published:** 2025-06-21

**Authors:** Mikhail Blagonravov, Anna Ryabinina, Ruslan Karpov, Vera Ovechkina, Maxim Filatov, Yulia Silaeva, Sergei Syatkin, Enzo Agostinelli, Vsevolod Belousov, Andrey Mozhaev

**Affiliations:** 1Institute of Medicine, RUDN University, 6 Miklukho-Maklaya St, Moscow 117198, Russia; ryabinina-ayu@rudn.ru (A.R.); syatkin-sp@rudn.ru (S.S.); 2Laboratory of Molecular Technologies, Shemyakin-Ovchinnikov Institute of Bioorganic Chemistry, Russian Academy of Sciences, Moscow 117997, Russia; kruslan148@gmail.com (R.K.); vs_ovechkina@mail.ru (V.O.); belousov@fccps.ru (V.B.); a.a.mozhaev@gmail.com (A.M.); 3Institute of Translational Medicine, Pirogov Russian National Research Medical University, Moscow 117997, Russia; 4Institute of Gene Biology, Russian Academy of Sciences, Moscow 119334, Russia; maxfilat@yandex.ru (M.F.); yulya.silaeva@gmail.com (Y.S.); 5Institute for Regenerative Medicine, Sechenov University, Moscow 119991, Russia; 6Department of Sensory Organs, Faculty of Medicine and Dentistry, Sapienza University of Rome, University Hospital Policlinico Umberto I, I 00161 Rome, Italy; enzo.agostinelli@uniroma1.it; 7International Polyamines Foundation, ETS ONLUS, I 00159 Rome, Italy; 8Federal Center of Brain Research and Neurotechnologies, Federal Medical Biological Agency, Moscow 117513, Russia; 9Life Improvement by Future Technologies (LIFT) Center, Moscow 143025, Russia; 10Joint Department with RAS Shemyakin-Ovchinnikov Institute of Bioorganic Chemistry, Faculty of Biology and Biotechnology, National Research University Higher School of Economics, Moscow 105066, Russia

**Keywords:** genetic models, transgenic, knockout, hypertension, cardiomyopathy, atherosclerosis, arrhythmias, adeno-associated dependoparvovirus A, CRISPR-associated protein 9

## Abstract

This review critically examines the evolving landscape of genetic animal models for investigating cardiovascular diseases (CVDs). We analyze established models, including spontaneously hypertensive rats, Watanabe hyperlipidemic rabbits, etc., and transgenic models that have advanced our understanding of essential and secondary hypertension, atherosclerosis, and non-ischemic diseases of the heart. This review systematically evaluates the translational strengths and physiological limitations of these approaches across species barriers. Particular attention is paid to emerging technologies—AAV-mediated gene delivery, CRISPR-Cas9 editing, and chemogenetic tools—that enable unprecedented precision in manipulating cardiac-specific gene expression to study pathophysiological mechanisms. We address persistent challenges including off-target effects and transgene expression variability, while highlighting innovations in synthetic vectors and tissue-specific targeting strategies. This synthesis underscores how evolving genetic technologies are revolutionizing cardiovascular research paradigms, offering refined disease models and optimized therapeutic interventions that pave the way toward personalized medicine approaches for the world’s leading cause of mortality.

## 1. Introduction

Animal models for investigating the mechanisms of pathogenesis underlying cardiovascular diseases can be categorized into three distinct groups: (1) models carried out using surgical intervention; (2) models in which a pathological process is initiated by introducing chemical or infectious agents into the animal body; (3) genetic models. Definitely, any combination of these methods for modeling a certain pathology are also possible. Each of these model groups has its advantages and limitations. A critical criterion for their selection is the degree to which the physiology of an animal’s circulatory system aligns with human physiology [1]. Unfortunately, none of the currently used models of cardiac diseases can entirely replicate the pathogenesis of the corresponding human pathology, although they do possess certain pros and cons [2,3,4,5,6]. It is also very important to use animal models providing a reliable means of evaluating the efficacy of pharmaceuticals and other methods of intervention in pathological processes in basic and preclinical investigations [2,7]. Recently, the horizons of cardiac pathology modeling have been significantly expanded through the employment of genetic instruments. The advent of extensive genetic resources has also opened new avenues for studying the activity of signaling pathways and modifying the functions of genes across various cellular types [8]. Genetic models are widely used in the investigations of different forms of cardiomyopathy, as they effectively reproduce the intricate processes underlying the disease, which is predominantly inherited. In particular, there are animal models of idiopathic cardiomyopathy, including dilated [9,10,11], hypertrophic [12,13], restrictive [14,15], and arrhythmogenic right ventricular dysplasia (cardiomyopathy) [16,17].

When discussing disorders of the cardiovascular system that can only be attributed to hereditary predispositions, such as hypertension, coronary artery disease (CAD), or various types of cardiac arrhythmia, genetic models provide a valuable tool for exploring the underlying mechanisms of these conditions.

At present, small mammal studies in the field of cardiac pathophysiology, such as mice and rats, are particularly prevalent. The advantages of using rodents in research include a short breeding cycle [18], the ability to perform genetic manipulations, and a wide range of available antibodies specific to mice, which can be used for both basic and pre-clinical in vivo studies [19]. The limitations concerning experiments on small animals include the technical difficulties associated with measuring various indicators using instrumental methods. It is also rather challenging to obtain enough biological material from such animals for research purposes (especially fluids, such as urine and blood samples). Nonetheless, advanced surgical and imaging techniques have alleviated at least some of these challenges [20].

The entire variety of genetic models of heart pathology that exist today can be subdivided into the following groups: (1) genetic strains selected to maintain phenotypic characteristics resulting from spontaneous genetic mutations and naturally occurring pathology; (2) genetically modified animals with certain genes knocked out; (3) transgenic organisms with the genome modified by the insertion of genes from unrelated species [21]; (4) models developed based on genetically encoded tools (vectors) [22,23,24,25].

When developing an experimental design, researchers commonly face the challenge of selecting an animal model that most closely resembles the actual pathological process observed in humans. In our review, we analyze genetic models of the most common cardiovascular diseases, focusing on their advantages, limitations, and the suitability of their application depending on the specific research task.

The process of preparing this review involved employing several methodical approaches in the search and selection of literature sources. The search was performed using databases, including, first of all, PubMed, Google Scholar, and Research Gate. When searching for publications, the following keywords were used (both separately and in combination) in general: genetic models; animal models, mice, murine, rat, rabbit, cat, feline, dog, zebrafish, transgenic; knockout; hypertension; cardiomyopathy; atherosclerosis; arrhythmias; adeno-associated dependoparvovirus A; CRISPR-associated protein 9. Depending on the particular section, we employed the following terms for our search: Section 2 —genetic models, animal models, inbred strains, heart, cardiac, hypertension, cardiomyopathy, atherosclerosis, coronary heart disease; Section 3—genetic models, animal models, transgenic, knockout, heart, cardiac, CRISPR/Cas9; cardiac arrhythmias, channelopathies, cardiomyopathy, atherosclerosis, myocardial hypertrophy; Section 4—genetic models, heart, cardiac, genetically encoded tolls, adeno-associated dependoparvovirus A, transduction, knockout, Cre-LoxP system, plasmid. Priority was given to publications from the last 5–10 years. In total, more than 900 sources were analyzed, and 244 of them were included in the final list of references for this review. The inclusion criteria were as follows: publications strictly related to the topic of this review (genetic models of cardiovascular disease), sufficient breadth and depth of coverage, relevance and reliability of sources (only articles from scientific journals and monographs were considered), as well as accessibility to the full-text version. And we also applied the following exclusion criteria: irrelevant sources, publications with methodological flaws, and sources that did not correspond to the methodology we were analyzing (we were interested in experimental research-based publications).

## 2. Genetic Strains Selected to Maintain Phenotypic Characteristics Resulting from Spontaneous Genetic Mutations and Naturally Occurring Pathology

Among the genetic animal models of cardiovascular pathology, inbred strains created by breeding animals with spontaneous mutations were developed earlier than other models—in the 1950s. These models, widely used in experimental work, have demonstrated their efficacy, as the corresponding phenotypic traits are largely reproduced in the majority of progeny.

The genetic models demonstrating cardiovascular pathology phenotype, the mechanisms of its formation, possible applications, their advantages and limitations, physiological relevance, genetic manipulability, reproducibility, and translational relevance are summarized in Table 1.

Animal models of hypertension: Today, the most common object for the study of hypertension is represented by rats. Most rats genetically predisposed to hypertension were derived from outbred Wistar and Sprague Dawley (SD) stocks [19]. Inbred strains that are known to reproduce this pathology are as follows: SHRs (spontaneously hypertensive rats), DSS (Dahl salt-sensitive) rats, FHH (fawn-hooded hypertensive) rats, the Milan hypertensive strain, the Lyon hypertensive rat, the Sabra hypertensive rat, Buffalo (BUF) rats, Goto-Kakizaki (GK) rats, Munich Wistar Frömter (MWF) rats, the New Zeland Genetically Hypertensive (NZGH) strain, and some more rare strains.

The SHR strain is the most used rat model that demonstrates a hypertensive phenotype, and it is widely used to study essential hypertension. This strain was developed by crossing outbred Wistar rats that had spontaneously increased blood pressure (BP) [26]. It is considered that the primary pathogenic mechanism underlying the onset of arterial hypertension (AH) in this strain of animals is neurogenic [27,28]. SHR animals also exhibit overexpression of the renin gene [29] and the presence of four mutations in the first 1100 base pairs of the first intron of the renin gene [30]. Thus, this model closely resembles the pathogenesis of human essential hypertension [28]. In adult male SHRs, SBP is typically above 180 mmHg and DBP is above 130 mmHg at rest, as established by the method of telemetric monitoring that eliminates the influence of stress or any other external factors [31,32].

In most cases, SHRs develop cerebral stroke, so this strain or its substrains (SHR-SP—spontaneously hypertensive stroke-prone rats) are widely used as a model for studying cerebral circulatory disorders [33,34]. This model can also serve as a valuable tool for investigating vascular and renal abnormalities in hypertension, as well as genetic mechanisms underlying its development. Moreover, SHRs can also be used to study the transition from compensatory left ventricular hypertrophy to heart failure [35,36].

The DSS (Dahl salt-sensitive) strain was developed by Lewis Dahl in the 1950s, and it represents inbred rodents that are extremely sensitive to high-salt diets [37,38]. Initially, the animals were fed on this diet, and it was found that some of them developed AH, while the others maintained a normal BP pressure level. After three generations, a strain of rats sensitive to high-salt diets was selected [39]. Interestingly, this study, on the one hand, confirmed the role of excessive salt intake as a risk factor for hypertension. On the other hand, it was shown that the development of hypertension due to salt intake is influenced by genetic predisposition. Subsequently, it was found that salt-sensitive rats have the allele of the gene encoding 11B-hydroxylase (*CYP11B1*), an enzyme necessary for the synthesis of a steroid that stimulates salt retention, which is responsible for the development of hypertension [40]. This model is particularly valuable for studying the role of sodium in the development of hypertension.

The FHH (fawn-hooded hypertensive) is an inbred strain that was selected by crossing outbred “Fawn-hooded” animals with chronic renal failure accompanied by a mild increase in BP, compared to Wistar rats [41]. In fact, the FHH strain is a genetic model of chronic renal disease, characterized by the development of glomerulosclerosis and proteinuria [42] followed by the development of hypertension [43]. It is still difficult to determine with certainty whether hypertension or nephropathy is the primary condition in this case. However, it was shown that in FHH K572Q rats, the *ADD3* gene contributes to modifying the myogenic response and the autoregulation of renal and cerebral blood flow, which results in increased susceptibility to kidney disease caused by hypertension [44,45].

The FH strain finds its application in the study of hypertensive nephropathy, as well as the genetic basis of hypertension.

The MHS (Milan hypertensive strain) was developed as a result of breeding Wistar rats that exhibited increased BP [46]. The main mechanism underlying the development of hypertension in animals from this strain is a genetically determined kidney dysfunction, characterized by an imbalance between the glomerular filtration and the tubular reabsorption of sodium and water [27]. The hypertension in this rat strain is linked to a mutation in the *ADD* gene, encoding the protein adducin (actin-binding protein) [44]. The mutation in this gene is also associated with renal dysfunction and cognitive disorders [47]. This model can be used to study primary hypertension and the genetic mechanisms of its development.

The Lyon hypertensive rat is an inbred strain of animals characterized by elevated BP levels. This strain was developed in the 1970s by a group of French researchers by repeated crossbreeding of Sprague Dawley rats with different BP levels, which resulted in the creation of three animal strains: those with normal BP (LN), those with elevated BP (LH), and those with lowered BP (LL) [48,49]. Interestingly, this model is not only characterized by signs of spontaneous hypertension but also demonstrates manifestations of metabolic syndrome [50], salt-sensitive hypertension [51], impaired renal function and proteinuria [52], and increased insulin blood levels and insulin-to-glucose ratio (insulin resistance) [50]. The results of genetic mapping in Lyon hypertensive rats showed that chromosome 17 contains two QTL clusters: one associated with body weight and organ hypertrophy, and the other associated with BP level, plasma lipids, and insulin levels [53]. It has recently been shown that a mutation in the gene *Ercc6 l2* appears to be the best candidate determining the phenotype of obesity and metabolism in a novel congenic strain LH17LNa [54]. For today, the Lyon hypertensive rat is the gold standard model for studying hypertension as part of metabolic syndrome.

Sabra hypertensive rats are a less commonly used genetic model for hypertension. Nowadays, the PubMed database contains a modest number of a bit over 90 articles. Two inbred substrains, Sabra hypertensive-prone (SHP) and Sabra hypertensive-resistant (SHR) rats, were selected: the former one exhibited increased sensitivity to salt with the development of hypertension [55]. Animals of both genetic substrains receiving an ordinary diet are characterized by normal BP [56]. The main mechanism of hypertension in this model is genetically determined salt susceptibility [57]. It is noted that chromosome 1 loci are associated with increased BP in this rat line [58,59]. Moreover, three potential gene loci of chromosomes 1 and 17, which play an important role in salt sensitivity and/or resistance, have been identified [59]. SBH animals also have proteinuria and nephrosclerosis [60]. The SBH strain is a good model for the study of salt-induced hypertension, as well as hypertensive nephropathy.

The New Zealand Genetically Hypertensive (NZGH) rat strain was developed in the 1970s by crossing outbred animals with a hypertensive phenotype [61]. It was shown that the renin–angiotensin–aldosterone and sympathoadrenal systems do not play a primary role in the development of hypertension in this animal model. However, they are characterized by increased reactivity of peripheral vessels to various vasoconstrictor agents, which is due to the involvement of neurogenic and myogenic factors, as well as structural factors. It is also notable that animals of this strain do not develop hypertension even when kept on a high-salt diet [61].

Some reports suggest the possibility of hypertension in Munich Wistar Frömter (MWF) rats, which are a model of chronic kidney disease associated with a decrease in the number of superficial glomeruli [62,63]. At the same time, there is little evidence in the literature indicating the development of persistent hypertension in this animal strain. There are also limited data concerning the possible genetic basis for its formation. In this regard, it is challenging to recommend this strain as a hypertension model.

Genetic models of hypertension in large animals: Research on hypertension employing large animal models is currently less prevalent. However, in some cases, it is necessary. Large animals are preferable for experiments using various surgical techniques, as well as intraventricular manometry. Spontaneous increases in BP were found in turkeys [64], rabbits [65], pigs [66], and dogs [67,68]. Strains of New Zealand and Dutch white rabbits with spontaneous hypertension were developed using the method of selective breeding [19].

Animal models of idiopathic cardiomyopathy: Idiopathic cardiomyopathies include dilated cardiomyopathy (DCMP), hypertrophic cardiomyopathy (HCMP), restrictive cardiomyopathy (RCMP), arrhythmogenic right ventricular dysplasia (cardiomyopathy), and unclassified cardiomyopathy. Currently, there are animal genetic models for every type of cardiomyopathy that reproduce the pathogenesis of these diseases.

Genetic models of dilated cardiomyopathy (DCMP): The most prevalent DCMP model involves Syrian golden hamsters derived from various genetic substrains of the BIO strain (BIO 1.50, BIO T0-2 and others) (Figure 1) [69,70].

This strain was first described in 1962. It was the result of breeding a naturally occurring mutant strain of Syrian hamsters, with an autosomal recessive mode of inheritance, with 100% penetrance. This strain exhibited signs characteristic of both cardiomyopathy and muscular dystrophy [69]. The main cause of structural and functional changes in the hearts of these hamsters is the deletion of the gene encoding delta-sarcoglycan. This protein is part of a complex associated with dystrophin. This leads to a loss in delta-sarcoglycan in the CMC [71]. The animal model of the BIO T0-2 substrain shows a complex of morphological changes, in particular, enlargement of the heart chambers and inflammatory infiltration of the myocardium [72]. Electron microscopy reveals shortening of sarcomeres in 7 out of 10 cases [70]. Moreover, in experiments on the same substrain, BIO T0-2, a complex of intra-cardiac and systemic hemodynamic abnormalities observed in human DCMP was identified: low cardiac output, increased preload, and decreased renal blood flow [73], as well as pronounced changes in the Frank–Starling mechanism [70]. All these facts allow us to recommend Syrian golden hamsters from the BIO strain as an optimal model for studying the pathogenesis and genetic basis of DCMP, as well as its gene therapy.

DCMP also occurs in wild and domestic turkeys and dogs (Figure 2) [74].

In domestic turkeys, in 2–5% of cases, DCMP develops within the first 4 weeks of life and occurs in the absence of other cardiac abnormalities. This condition is characterized by dilated cardiomyopathy, leading to heart failure and high mortality rates in affected turkeys. The underlying causes are not fully understood but may involve genetic predisposition and environmental factors [75]. In turkeys with DCMP, low-molecular cardiac troponin T (cTnT) is expressed in myofibrils due to the absence of exon 8. This is accompanied by a change in the conformation and binding affinity for TnI и TP [76]. We should note that the presence of DCMP in turkeys and dogs has not resulted in the development of genetic strains with the corresponding pathological phenotype. In this regard, we are not discussing the existence of a specific animal model. At the same time, mutations in the troponin T gene in these animals allow us to study the genetic mechanisms of DCMP, as it is known that troponin T mutations are found in 3–6% of human DCMP patients [77].

Genetic animal models of hypertrophic cardiomyopathy (HCMP). The estimated incidence of hypertrophic cardiomyopathy (HCMP) in the general population is approximately 1:500 [78]. In contrast to DCMP, familial cases of HCMP are 30–50% [79,80], while only 30–40% of them can be attributed to a genetic cause [79,81]. HCMP has a well-established genetic basis (mutation) [82,83,84]. Most patients have mutations in the gene *MYH7* (OMIM: 160760) encoding myosin heavy chains and the gene *MYBPC3* (OMIM: 600958) encoding myosin binding protein, as well as cardiac troponin C (*TNNC1*; OMIM: 191040), cardiac troponin I (*TNNI3*; OMIM:191044), cardiac troponin T (*TNNT2*; OMIM:191045), cardiac actin (*ACTC1*; OMIM: 102540), α-tropomyosin (*TPM1*; OMIM:191010), regulatory myosin light chain, essential myosin light chain, and titin/connectin. In more than 50% of cases, the disease is caused by mutations in the *MYH7* or *MYBPC3* genes [85]. In 5–10% of cases, the disease is caused by a mutation in the genes encoding non-sarcomeric proteins, which are associated with neuromuscular diseases [86].

Naturally occurring HCMP is often found in pet cats, rendering them as an acceptable model of this disease [87]. The prevalence of HCMP in the general cat population is 10 to 15% [88]. An even higher risk is typical for some certain breeds of felines, including the Maine Coon cat, Persian, Ragdoll, and Sphynx [87]. It is important that feline HCMP is characterized by the same morphological, pathogenetic, and clinical features as human HCMP [89,90]. As in humans, the pathogenesis of feline HCMP is based on mutations in the genes that encode sarcomere proteins [87]. Maine Coon cats have mutation A31P in the gene *MYOC*, which is found in 22–42% of individuals. In Ragdoll animals, the mutation R820W of the same gene is observed in 27% of cases [91]. The R820W mutation of the *MYBPC3* gene has also been found in human patients with HCMP, which allows us to consider this genetic abnormality as a common pathogenic basis for HCMP in both cats and humans [92]. Nonetheless, not all Maine Coon cats with advanced HCMP have the A31P mutation in the gene *MYBPC3* [87]. Despite the existence of animals with naturally occurring HCMP, there is currently a wide number of transgenic models. The disadvantage of rodent transgenic models is the fact that, even with the same genetic mutations as in humans, the phenotypic manifestations can be different. Mice with mutations in the gene encoding myosin-binding protein (MYBPC) do not develop myocardial hypertrophy [87].

Genetic models of restrictive cardiomyopathy (RCMP): In approximately 30% of cases, idiopathic RCMP has a familial character. In particular, in patients with familial RCMP, mutations associated with this disease were found in the following genes: *TNNT2* (cardiac troponin T; OMIM: 191045), *TNNI3* (cardiac troponin I; OMIM: 191044), *MYBPC3* (myosin-binding protein C), *MYH7* (myosin heavy chain 7; OMIM: 160760), *MYL2* (myosin light chain 2; OMIM: 160781), *MYL3* (myosin light chain 3; OMIM: 160790), *DES* (desmin; OMIM: 125660), *MYPN* (myopalladin; OMIM: 608517), *TTN* (titin; OMIM:188840), *BAG3* (Bcl-2-associated athanogene 3; OMIM: 603883), *DCBLD2* (discoidin, CUB and LCCL domain containing 2; OMIM: 608698), *LMNA* (lamin A/C; OMIM: 150330), and *FLNC* (filamin C; OMIM: 102565) [93]. And as in the case of HCMP, these are mutations in genes encoding structural proteins of sarcomeres or proteins associated with sarcomeres [94]. Some clinical and histopathological features of RCMP are like those of HCMP. Moreover, mutations were found in some of the same genes (*TNNT*, *TNNI3*, *MYBPC3*, *MYH7*). And we may suggest that DCMP and HCMP share a few common genetic and pathogenic mechanisms [95]. Naturally occurring genetically determined RCMP was found in cats. Genetic animal models with similar mutations in the genes of myosin, *TNNI3*, and *MYPN* demonstrate the diastolic dysfunction phenotype, which is a major clinical manifestation of human RCMP [15]. Particularly, under natural RCMP in cats, there is a pronounced thickening of the endocardium, as well as replacement fibrosis and deformation of the heart chambers [96,97].

The use of animal models for HCMP and RCMP is complicated by the fact that inbred strains have not been developed. Since, as mentioned previously (Figure 3), the pathogenic, morphological, and clinical patterns of these diseases more closely resemble human cardiomyopathy than transgenic models, it is a more suitable approach for translational studies. In this regard, it seems reasonable to use an approach that involves collaboration with veterinary clinics and relevant services that are ready to provide autopsy tissue samples. Such an approach was used in the study performed by Kimura Y. et al. [97].

Animal models of arrhythmogenic right ventricular dysplasia/cardiomyopathy (ARVC): ARVC is a rare genetically determined disease that is inherited in an autosomal dominant pattern. It is characterized by the replacement of myocardial elements with fibrous and fatty tissue, which can lead to heart failure and sudden cardiac death. This disease is associated with mutations in the genes encoding proteins of the desmosome complex [98], in particular, *DSP* (desmoplakin; OMIM: 125647) [99], *PKP2* (plakophilin 2; OMIM: 602861) [100], *DSG2* (desmoglein 2, desmocollin 2; OMIM: 125671) [101], *JUP* (plakoglobin (related to γ-catenin; OMIM: 173325) [102,103], *TMEM43* (transmembrane protein 43; OMIM: 612048), and *ILK* (integrin-linked kinase; OMIM: 602366) [104,105].

Mutations in the gene *PKP2* are the most common cause of familial ARVC [106]. Today, there are no available animal models of ARVC developing because of mutations in the gene *DSP* [98]. However, there are knockout murine models of deficiency of the genes *PKP2* [107], *DSG2* [108], and *JUP* [109], as well as DSC2-related ARVC in zebrafish [110]. Moreover, it was also found that overexpression of wild-type and p.P111L-mutant *TMEM43* in a transgenic zebrafish model was associated with cardiac enlargement and cardiomyocyte hypertrophy, which confirms the role of transcriptomic alterations in the pathogenesis of *TMEM43*-associated cardiomyopathy [111]. It was also shown that two missense variants (p.H33N and p.H77Y) in the *ILK* gene are associated with arrhythmogenic cardiomyopathy. Additionally, expression of human wild-type and mutant *ILK* in zebrafish suggests that p.H77Y and p.P70L may be responsible for cardiac dysfunction and death within 2–3 weeks after birth [105]. Transgenic models will be discussed in more detail in the following section of this review. As regards natural genetic models, spontaneous ARVC is observed in dogs and cats. In particular, spontaneous genetically inherited ARVC closely resembling this disease in humans was described in boxer dogs. It is characterized by ventricular arrhythmia, syncope, and sudden cardiac arrest. Morphological examination reveals a reduction of the number of cardiomyocytes and their replacement by fatty or fibrous–fatty elements [112]. Spontaneous ARVC in pet cats was described as a model for this pathology in several studies.

Fox P.R. et al. examined 12 cats with this disease, and in 8 animals, they diagnosed right ventricular congestive heart failure with supraventricular and ventricular tachyarrhythmias. Morphological analysis demonstrated a marked enlargement of the right ventricle and thinning of its wall in all 12 animals, atrophy and death of CMCs, replacement fibrosis, and fatty degeneration [113] (Figure 4).

Genetic animal models of atherosclerosis and coronary heart disease: Currently, surgical occlusion (ligation) of one of the branches of the coronary artery is still successfully used as a model for acute myocardial infarction. Obviously, this approach does not reproduce the natural pathogenesis of coronary heart disease, which is typically characterized by a slow progressive atherosclerotic lesion of muscular–elastic arteries. The classical model of atherosclerosis, suggested by N.N. Anichkov, which has been used for many decades, also has certain limitations. In the 1980s, Watanabe Y. et al. used selective breeding to develop a genetic strain of rabbits with congenital hypercholesterolemia, Watanabe heritable hyperlipidemic (WHHL) rabbits, as a model for atherosclerosis [114]. These animals have a genetic abnormality resulting in a disorder of the expression of cellular receptors to LDL, similar to human familial hypercholesterolemia [115]. Rabbits of the WHHL strain are characterized by increased LDL levels, followed by the development of severe atherosclerosis. Atherosclerotic arterial damage undergoes several stages in its development. As the intimal damage progresses, deposits of lipids and foam cells appear on the inner surface of the vessel, leading to the formation of plaques and an increase in lipid content in the tunica media. Afterwards, calcification often takes place [115,116]. The atherosclerotic changes in arteries in this case are more similar to the morphogenesis of atherosclerosis in humans compared to the cholesterol-fed rabbit model [117].

As concerns the use of WHHL strain rabbits as a model for coronary heart disease, including myocardial infarction, there is still no clear consensus on this matter. Some studies provide data on the development of coronary syndrome and spontaneous myocardial infarction following severe atherosclerosis in animals of this strain [118,119]. However, we could not find confirmation of these observations in the works of other authors. At the same time, the WHHL rabbit model is widely used to study the cholesterol-lowering effects of statins [120,121,122,123] and non-statin compounds [124,125] in atherosclerosis. Animals from the WHHL family have also been successfully used in the development of gene therapy for atherosclerosis [126]. Particularly, the transfection of the *LDLR* (low-density lipoprotein receptor; OMIM: 606945) gene into the liver cells of WHHL rabbits using a lentiviral vector resulted in a significant decrease in serum cholesterol levels [127]. There are also transgenic models of hypercholesterolemia and atherosclerosis developed in mice and rats. However, it is rabbits that have several metabolic features close to humans [128]. Rabbits have high activity of the cholesterol ester transporter protein (CETP) in blood plasma [129].

Summarizing the above, we can conclude that the WHHL strain is the optimal model of atherosclerosis that can be successfully used to investigate its pathogenetic features, as well as to develop new methods of pharmacotherapy and gene therapy. However, to induce ischemic and ischemic–reperfusion injury to the myocardium, models based on the mechanical occlusion of coronary arteries are more appropriate.

## 3. Transgenic and Knockout Animal Models

Methods of genome modification in animal models of cardiovascular pathology. Several genome modification methods are commonly used to create animal models. The oldest one is transgenesis by random integration. This method is based on the ability of DNA molecules delivered into the zygote pronucleus to integrate into the genome [130]. The design of genetic constructs for random integration necessarily includes a gene of interest, a promoter, and regulatory elements to mitigate potential position effects. The integration of delivered constructs occurs randomly, making it impossible to predict the insertion site, the number of integrated transgene copies, or the level of their expression in advance. This phenomenon is known as the “position effect of the transgene” [131,132]. Therefore, when creating transgenic animal strains using the random integration method, multiple strains with the same transgene insertion are typically generated. In this case, a strain with the desired level of transgene expression is subsequently selected. For example, a mouse model of short QT syndrome was generated using the random transgenic insertion method [133]. Similarly, a model of myocardial hypertrophy induced by the overexpression of platelet-derived growth factors (PDGFs) has also been developed using this approach [134]. Moreover, transgenic mice with overexpression of desmocollin-2 serve as a model for cardiomyopathy [135].

The development of the CRISPR/Cas9 genome editing system has significantly increased the efficiency of gene modification. This method is widely used for generating knockout models, particularly in research focused on evaluating the role of genes in cardiac function [136,137].

Another commonly used approach for obtaining genetically modified animal strains with desired mutations is the generation of chimeric embryos via embryonic stem cell (ESC) microinjection. In this method, the genome of ESCs is modified, and clones with the desired genetic modifications are selected and used to create chimeric embryos. When such chimeric mice (F0) are born, genetically modified non-chimeric offspring (F1) must be obtained from them. To produce homozygous animals, it is necessary to breed heterozygous F1 individuals. This process requires a significant period, but it enables the creation of animals with highly precise genetic mutations. The ESC method for generating genetically modified animals is widely used in various fields, particularly in cardiovascular research [138,139].

Recent advances in genome sequencing have enabled the identification of the genetic basis of many myocardial pathologies, including channelopathies and cardiomyopathies [140,141,142,143,144]. At the same time, genome editing technologies empower the development of near-personalized models replicating mutations observed in specific patients (or patient groups) [145,146]. These models facilitate the investigation of various factors, such as the role of pro-inflammatory responses in cardiovascular diseases [147]. This raises the following question [148,149,150]: are certain laboratory animals appropriate as model organisms for specific studies?

Common genetically modified model animals of cardiovascular pathologies: Research has extensively explored animal models of heart conditions, and mice remain the predominant choice for modeling these diseases [1,148,149,150,151,152,153]. Their advantages include a well-characterized genome homologous to the human one, established genetic modification techniques, and relatively low operational costs. Genetically modified mice have been developed to mimic various human cardiac conditions, such as cardiomyopathies, e.g., transgenic mouse models of dilated cardiomyopathy carrying the TTNtv c.13254T > G mutation [154] or Lmna c.1621C > T mutation [155], transgenic murine models of hypertrophic cardiomyopathy carrying the c.1826dupA mutation [156] or p.G823E mutation [157]; arrhythmias, e.g., mice models of Mog1 knockout [158], Kcne3 knockout [159], or the p.Asn1768Asp mutation [160]; and other inherited heart diseases, e.g., mice models of congenital heart defects, including truncating FOXJ1 variant (c.784_799dup; p.Glu267Glyfs*12) [161], Fgf10 and Fgfr2-IIIb [162], and Hnrnpa1 mutations [163]. However, there are some challenges in using mice as biomedical models. In some cases, mutations that are homologous to the ones found in humans are not realized as a pathological phenotype in mice or result in a different phenotype [164]. Moreover, the investigation of cardiac development during embryogenesis is complicated due to difficulties in dynamic imaging, despite existing ex utero culturing techniques [165,166].

Another popular model organism is the zebrafish (D. rerio) [17,167,168]. Homozygous D. rerio with a mutation in the *bag^3e2/e2^* gene serve as a model for dilated cardiomyopathy [169]. Knockout of gtpbp3 leads to hypertrophic cardiomyopathy [170]. Overexpression of a transgenic construct containing the atrial-specific myosin light chain 4 gene *MYL4* (OMIM: 160770) with the p.Glu11Lys mutation results in the development of arrhythmia [171]. One of its key advantages is the relatively high genomic homology with humans [172]. Additionally, D. rerio exhibits rapid embryonic development: its heart begins to beat just one day after fertilization [173]. The transparency of zebrafish embryos and their external development facilitate easy observation of embryonic stages. Another significant advantage of D. rerio as a model organism is the low cost of genetic modification [174] and the general simplicity of maintaining these animals. However, despite its many advantages, the major limitation of D. rerio for the research of cardiac disorders is its two-chambered heart. For this reason, certain processes characteristic of the human heart cannot be fully modeled using D. rerio.

Rabbits are an excellent model organism for numerous biomedical studies [175]. This is primarily due to the similarity between rabbit and human lipid metabolism, which enables the development of appropriate models of lipid metabolism disorders, including, first of all, atherosclerosis [176,177,178]. Moreover, the efficiency of the CRISPR/Cas9 genome editing system in rabbit embryos is exceptionally high, significantly accelerating and simplifying the creation of knockout models [176,179,180,181]. Despite the existence of rabbit models for heart diseases [182], the use of this species in biomedical research remains limited. This is primarily due to the relatively high cost of their maintenance and their relatively long reproductive cycle. Transgenic rabbits serve as models for monogenic human diseases. For example, the expression of a mutant variant of *β-MHC-Q403* leads to myocardial hypertrophy [183]. Additionally, the expression of a mutant variant *KCNQ1/KvLQT1-Y315S* (a dominant-negative mutation in the K^+^ channel α-subunit [KvLQT1]) in models prolonged QT/APD [184].

The comparative advantages and disadvantages of popular animal species used for modeling heart diseases are presented in Table 2.

Since there is currently a vast variety of transgenic animal models available for studying cardiovascular pathology, it would be challenging to thoroughly analyze all of them in the frame of single review. In this regard, we only present general principles and approaches that are used in this area, as well as certain examples of transgenic models of diseases of the cardiovascular system.

Transgenic animal models of genetic channelopathies and some types of cardiac arrhythmias:

Channelopathies arise from mutations in ion channel genes or their regulatory components, increasing the risk of arrhythmias in structurally normal hearts. Various animal models have been developed to investigate the pathophysiology of these disorders.

For example, mutations leading to spontaneous Ca^2^⁺ release from *RYR2* (ryanodine receptor 2; OMIM: 180902) sarcoplasmic reticulum (SR) Ca^2^⁺ release channels result in catecholaminergic polymorphic ventricular tachycardia (CPVT) and atrial fibrillation (AF). The *RyR2^+/^^−^* mouse model exhibits arrhythmogenic phenotypes like the ones that are observed in CPVT patients [185].

Cardiac calsequestrin 2 (*CASQ2*; OMIM: 114251) mutations, transmitted by an autosomal recessive type of inheritance, lead to a loss-of-function phenotype. *CASQ2* acts as a high-capacity Ca^2^⁺ buffer in the SR and regulates *RYR2* gating. Deletion of *CASQ2* in mice (*Casq2^−^^/^^−^)* results in severe exercise- and catecholamine-induced arrhythmias, mimicking the human disease phenotype [186].

Congenital long QT syndrome (LQTS) is often a multi-organ disorder caused by mutations in genes essential for cardiac repolarization, leading to the following symptoms: seizures, syncope, arrhythmia, and sudden cardiac death. The majority of LQTS cases arise from mutations in *KCNQ1* (potassium channel voltage-gated KQT-like subfamily member 1; OMIM: 607542), *KCNH2* (potassium channel voltage-gated subfamily H member 2; OMIM: 152427), or *SCN5A* (sodium voltage-gated channel alpha subunit 5; OMIM: 600163) [187]. The *Kcnq1^−^^/^^−^* mouse model exhibits characteristic features of LQTS, including abnormal T- and P-wave morphologies and prolonged QT and JT intervals when assessed in vivo [188]. AF can also be triggered by mutations in *KCNE1* (potassium channel voltage-gated Isk-related subfamily member 1; OMIM: 176261) and *SCN5A* with corresponding mouse models replicating AF susceptibility [189,190].

Brugada syndrome, characterized by ST-segment elevation, partial bundle branch block, arrhythmia, and sudden cardiac death, is most commonly associated with *SCN5A* mutations. The *Scn5a^+/^^−^* mouse model displays conduction disease phenotypes that parallel human Brugada syndrome [191].

Beyond these models, numerous studies have explored the effects of mutations, deletions, or dysregulation of ion channels, transcription factors, developmental pathways, and hormones to better understand the mechanisms underlying cardiac arrhythmias [1]. These animal models remain essential for elucidating disease mechanisms and evaluating potential therapeutic strategies.

Transgenic animal models of cardiomyopathy:

Dilated cardiomyopathy (DCMP): CRISPR/Cas9 genome editing of the dystrophin gene was used to create a Duchenne dystrophy animal model characterized by the development of heart pathology resembling DCMP [192]. Mutant rats developed left and right ventricular systolic dysfunction and myocardial fibrosis. The animals also exhibited systemic muscle atrophy accompanied by high mortality [193]. Several other genetic models of idiopathic DCMP have been created in mice. Homozygous mice with desmin knockout developed heart damage characteristic of DCMP, including thinning of the left ventricle, fibrosis, and systolic dysfunction observed in autosomal recessive desminopathies [194]. Progressive DCMP may be caused by a disruption of the nuclear envelope structure of cardiomyocytes, which may be due to a mutation in the *LEMD2* (LEM domain-containing protein 2; OMIM: 616312) gene [195]. The editing of desmosomal genes (desmocollin-2 and desmoglein-2) leads to the development of biventricular cardiomyopathy associated with inflammation, necrotic death of cardiomyocytes, and replacement sclerosis. This complex of changes closely resembles the pathogenesis of DCMP and can serve as its genetic model [135].

Hypertrophic cardiomyopathy (HCMP): For today, more than 1400 autosomal mutations transmitted by the autosomal dominant type of inheritance have been identified in 11 genes encoding proteins of the thick and thin strands of the sarcomere or components of neighboring Z-discs. Transgenic animal models include the following animal species: rodents, rabbits, and zebrafish [196]. The first animal model developed to study the pathogenesis of HCMP was a murine strain carrying a missense mutation in codon 403 in the *MYH6* gene corresponding to the R403Q mutation in *MYH7* (myosin heavy chain 7 cardiac muscle beta; OMIM: 160760) in humans. The R403Q model exhibited various pathological features, including myocellular disorder, fibrosis, and diastolic dysfunction, reproducing the pathology observed in human tissues [197]. Myosin-binding protein-C (MyBP-C) is the most common gene involved in the pathogenesis of HCMP in humans. A series of murine strains carrying modified MyBP-C with deleted myosin and titin-binding domains were generated and studied [198]: these transgenic mice developed mild hypertrophy, impaired sarcomere structure, and an altered ability to exercise [199]. Further investigations provided a wide range of transgenic models, predominantly in mice and other rodents, through the manipulation of specific involved genes.

Restrictive cardiomyopathy (RCMP): As it was previously noted, the pathogenesis of RCMP is linked to mutations in the myosin genes *TNNI3* and *MYPN*. In the early 2000s, a model of transgenic mice (cTnI193His) simulating cTnI R192H mutation (cTnI R193H in the mouse sequence) in the human heart was created. Another strain of transgenic mice with the RCM cTnI K178E mutation was also developed [200]. In cTnI K179E transgenic mice, a sharp enlargement of both atria was observed in the absence of ventricular hypertrophy and dilation. They exhibited hemodynamic features similar to cTnI193His mice, which confirms the development of RCMP as a result of the *TNNI3* mutation: the animals had severe cardiac dysfunction [201]. There was also a marked hypersensitivity of cardiomyocytes to Ca^2+^ [202].

Arrhythmogenic right ventricular dysplasia/cardiomyopathy (ARVC): Since most patients suffering from ARVC carry one or more genetic variations of desmosomal genes, the number of cardiac desmosomes is significantly reduced in embryos with Iup deficiency. Consequently, it was also shown that the knockout of Dsx, Ds2, and Psp2 leads to embryonic death in mice due to severe heart failure. [99,101,107]. Later, based on advances of animal genetic engineering, various transgenic and knockout animal models were created, including the strains with both desmosomal and non-desmosomal protein knockout [16].

Transgenic animal models of atherosclerosis:

Transgenic animal models have played a crucial role in elucidating the mechanisms of atherosclerosis and evaluating the efficiency of therapeutic interventions. Among rodents, mice remain the most widely used objects for modeling due to their genetic controllability, cost-effectiveness, and short disease induction period. Apolipoprotein E-deficient (apoE^−/−^) and low-density lipoprotein receptor-deficient (Ldlr^−/−^) mice are the cornerstone models, accounting for over 95% of atherosclerosis studies [203]. ApoE^−/−^ mice develop spontaneous hypercholesterolemia and aortic lesions on a chow diet, with accelerated plaque progression under high-fat diets, while Ldlr^−/−^ mice require dietary cholesterol to elevate LDL and form lesions [204,205,206]. However, murine models differ from humans in lipoprotein metabolism—mice lack cholesteryl ester transfer protein (CETP) and exhibit HDL-dominant profiles—limiting translational extrapolation.

To address these disparities, novel models have been developed: apoE^3^-Leiden. CETP mice incorporate human *CETP* (cholesterol ester transfer protein plasma; OMIM: 118470), enabling LDL-driven atherosclerosis and human-like drug responses [207]. Moreover, PCSK9 gain-of-function adeno-associated virus (AAV) models induce hypercholesterolemia without germline editing, offering flexibility for studying plaque calcification and regression [208]. Advanced models like SR-BI KO/ApoeR61^h/h^ mice replicate coronary atherosclerosis and spontaneous myocardial infarction, bridging the gap between murine physiology and human clinical outcomes [209].

Rabbit models, particularly Watanabe heritable hyperlipidemic (WHHL) and transgenic strains, offer unique advantages due to their human-like lipoprotein metabolism, including LDL dominance and CETP activity. WHHL rabbits, deficient in LDL receptors, develop spontaneous hypercholesterolemia and complex lesions resembling human plaques, making them valuable for studying plaque rupture and statin efficacy [210]. Transgenic rabbits expressing human lipoprotein(a) or apoB-100 further enable research on lipid-driven atherosclerosis and vascular calcification, complementing rodent models in translational studies [211].

Besides mice, hamsters and guinea pigs offer human-like lipoprotein profiles, including CETP activity and LDL dominance. CRISPR-generated Ldlr^−/−^ hamsters develop severe atherosclerosis with coronary lesions and thrombotic events under high-fat diets, mirroring human pathophysiology [212]. Rats, despite their natural resistance, show modest lesion formation in apoE^−/−^ or Ldlr^−/−^ models after prolonged dietary induction, but their utility remains limited compared to other rodents [213,214].

In conclusion, while apoE^−/−^ and Ldlr^−/−^ mice dominate in atherosclerosis research, creating models like CETP-transgenic mice, PCSK9-AAV systems, and CRISPR-engineered hamsters enhances translational relevance by recapitulating human lipid metabolism and complex plaque phenotypes. Species selection should align with experimental goals, balancing genetic manipulability, lipoprotein physiology, and clinical mimicry [211,212].

Transgenic animal models of myocardial hypertrophy:

Transgenic animal models are indispensable tools for studying the molecular mechanisms underlying myocardial hypertrophy, one of the frequently occurring features of cardiovascular diseases.

Mice overexpressing calcineurin develop robust hypertrophy and heart failure, mimicking pressure-overload phenotypes. This model has been pivotal in establishing the role of calcineurin–nuclear factor of activated T-cells (NFAT) signaling in pathological remodeling, with sustained activation leading to fibrosis, apoptosis, and adverse cardiac outcomes [215].

Conversely, transgenic overexpression of constitutively active Akt induces physiological hypertrophy, characterized by increased cardiomyocyte size without systolic dysfunction, highlighting the distinct pathways governing adaptive versus maladaptive hypertrophy. Investigation of cardiac-specific Akt overexpression in mice has demonstrated a remarkable increase in cardiac contractility and concentric left ventricular hypertrophy, with activation of the glycogen synthase kinase3-β/GATA4 pathway rather than MAPK signaling [216].

Recent advancements in genome-editing technologies, particularly CRISPR/Cas9, have revolutionized this field by enabling precise modeling of human hypertrophic cardiomyopathy (HCM)-associated mutations. For example, knock-in models of *MYH7* (β-myosin heavy chain) and *MYBPC3* (myosin-binding protein C3) mutations have provided critical insights into sarcomere dysfunction, impaired calcium handling, and metabolic dysregulation, closely reflecting human disease progression. Current research has expanded to explore targeted drugs like myosin ATPase inhibitors for treating these genetic variants in HCM [217].

Particularly, *MYBPC3* truncations impair diastolic function, while *MYH7* mutations promote hypercontractility and energy depletion, offering mechanistic clarity on genotype-phenotype correlations. Some studies have shown that stepwise loss of cMyBPC results in reciprocal augmentation of myosin contractility, with important implications for therapeutic strategies targeting myosin activity [217].

Transgenic models with mutations in the human ventricular myosin essential light chain (ELC) are especially informative for differentiation between pathological (as in Tg-A57G mice) and physiological hypertrophy (as in Tg-Δ43 mice) [218].

Despite their utility, transgenic models face certain limitations, including interspecies differences in calcium handling (e.g., reduced sarcoplasmic reticulum Ca^2^⁺-ATPase [SERCA2a] activity in rodents) and β-adrenergic signaling, which complicate translational research. Interestingly, transgenic expression of SERCA2a in mice exposed to aortic stenosis has shown the enhancement of survival rate and maintenance of contractile function during the transition from adaptive hypertrophy to early heart failure, highlighting the critical role of defective SR Ca^2^⁺ function in pressure overload-induced heart failure [219].

To address the multifactorial nature of hypertrophy, hybrid models combining transgenics with pressure-overload (e.g., transverse aortic constriction) or metabolic stress (e.g., high-fat diet) are increasingly utilized [220].

## 4. Genetically Encoded Tools in Cardiovascular Disease Modeling

Advantages of using animal models of the cardiovascular system with genetically encoded tools: Currently, the use of genetically encoded tools is actively considered in the context of gene therapy for various cardiovascular diseases, such as hereditary and acquired cardiomyopathies, atherosclerosis, arterial hypertension, heart failure, and others [221]. However, in biomedical research, there are also some approaches to create animal models with specific genetic modifications that cause pathological features of certain cardiac diseases.

There are many animal models of cardiovascular diseases produced by microinjection of plasmid and transposon sequences into the zygote to breed transgenic strains that are characterized by stable, intergenerational changes in the genotype throughout their ontogeny (knockout strains, strains overexpressing the target gene, etc.) [222,223,224]. To study socially significant acquired cardiovascular diseases associated with lifestyle, environmental conditions, age, drugs, and other exogenous factors, such models can be considered non-valid, since the specified changes in the genome of animals can affect the embryogenesis of many systems and organs, as well as the expression of other non-target genes during postembryonic development of the organism. They may also cause pathological changes in the heart and blood vessels without controlled experimental conditions [220].

For example, transgenic mice (DCM-TG9) used to model dilated cardiomyopathy (DCMP) with reduced or increased cardiac-specific activity of the enzyme phosphoinositide-3-kinase (PI3K) have already demonstrated changes in cardiac systolic function, remodeling of ventricular configuration, and premature death at a young age, which may complicate the study of acquired forms of cardiomyopathy in such models [225]. Mice with knockout of apolipoprotein E and scavenger receptor class B type I genes used to model atherosclerosis are characterized by frequent undesirable complications in the form of spontaneous plaque ruptures with subsequent infarction in target organs [220].

Some transgenic models require the administration of additional substances for induction or knockout of the target gene, so in mice with TRE-PKM1 and α-MHC-tTA transgenic constructs, it was required to provide animals with regular administration of doxycycline with drinking water, including during the breeding, gestation, and feeding periods, to achieve cardiac-specific induction of the muscle isoenzyme pyruvate kinase 1 (PKM1) at the time required by the investigators; this approach may also cause systemic, undesirable, and non-selective effects on functional, molecular, and morphological parameters in the study of cardiovascular disease pathogenesis [226].

Another popular approach in the creation of transgenic disease models is the use of the Cre-LoxP system, such as lifetime gene knockout induced by tamoxifen administration, which enables activation of target pathways in the adult animals, bringing researchers closer to real conditions for modeling common cardiovascular diseases [227,228]. However, numerous non-specific toxic effects of the Cre-LoxP system are known, complicating the study of specific etiological factors in the development of particular cardiovascular diseases: toxicity to cardiomyocytes, endotheliocytes, blood cells, caused by DNA damage, chromosomal abnormalities, disruption of cell proliferation cycles, and the mechanism of cell death [229].

In summary, the use of genetically encoded tools delivered to wild-type (WT) animals at a certain age, combined with other experimental conditions—surgical models, diets, drug administration, and others—could be considered to model the most common cardiovascular diseases associated with patients’ lifestyle characteristics.

Specific examples of animal models of cardiovascular disease using genetically encoded tools: To create representative animal models of cardiovascular pathologies using genetically encoded tools (e.g., plasmids), it is necessary to build a genetic construct functioning under an organ-specific promoter, select the right serotype of adeno-associated viral particles (AAV) for cardiac-specific plasmid delivery, and choose an adequate method to introduce AAV into the animal in vivo. Thus, depending on the purpose of the study, the biological species of the animal, and the selected virus serotype, there are many different surgical methods for in vivo administration of viral particles containing the target genetically encoded tool for delivery to the heart: intravenous injection, intracoronary introduction by selective retroinfusion, intramyocardial (intramyocardial), intra-aortic injection, and others [22,230]. To enhance cardiomyocyte transduction in vivo, depending on the selected AAV serotype, there are some approaches that enable to modify animal models, such as induction of tissue-specific overexpression of the multi-serotype AAVR receptor or additional administration of recombinant human vascular endothelial growth factor (VEGF) [22,231].

Increased expression of calcium-binding protein (HRC) in the sarcoplasmic reticulum is one of the causes of myocardial hypertrophy development, with subsequent progression of chronic heart failure due to disruption of the cytosolic Ca^2+^ transport cycle in cardiomyocytes. In a recent study, an attempt to eliminate HRC overexpression, an AAV9-mediated genetically encoded system for HRC gene knockdown (HRC-KD) was proposed. AAV9 was delivered by injection into the WT tail vein of 10-month-old C57BL/6 mice with transverse aortic constriction surgery to simulate heart failure. The control group comprised the operated mice that were not subjected to HRC knockdown. Lifetime inhibition of HRC gene expression resulted in an enhanced sarcoplasmic Ca^2+^ leak into the cytoplasm of cardiomyocytes by increasing phosphorylated forms of ryanodine receptor 2 (RYR2) and phospholamban (PLB) compared with controls. In turn, the increased cytosolic Ca^2+^ concentration activated the mitochondrial pathway of apoptosis through phosphorylation of Ca^2+^/calmodulin-dependent protein kinase II (CaMKII) and p38 mitogen-activated protein kinase (p38 MAPK) proteins, and the level of phosphorylated CaMKII was 2.8-fold higher compared to the control group. The increased cleaved form of caspase-3 and Bax/Bcl-2 ratio analyzed by TUNEL assay on myocardial histological sections also confirmed the enhanced development of heart failure in HRC-KD mice with transverse aortic constriction. According to the results of echocardiography in HRC-KD mice 11 weeks after the operative period, there were signs of heart failure aggravation, including decreased thickness of the posterior wall of the ventricles and interventricular septum combined with increased ventricular dilatation. Moreover, histological examination revealed pronounced fibrosis. So, the described transverse aortic constriction surgery combined with HRC gene knockdown performed with genetically encoded tools may be used as a model of chronic heart failure progression following acquired cardiovascular diseases [232].

To study the pathogenesis of autosomal dominant Brugada syndrome characterized by the R104W mutation in the *SCN5A* gene, a model using genetically encoded tools in C57BL/6 mice was developed. This mutation causes dysfunction of the alpha-subunit of the cardiac Na^+^-channel, Na_v_1.5; the mutant form of Na_v_1.5-R104W can interact with WT channels, causing a decrease in Na^+^ current in the sarcoplasmic reticulum of cardiomyocytes, which is responsible for the development of ventricular fibrillation followed by possible sudden cardiac death. To generate such models, newborn mice were cotransduced via the jugular vein of AAV9 with the plasmids AAV-cTnT-5′hNa_v_1.5-R104W (control mice were injected with AAV-cTnT-5′hNa_v_1.5-WT) and AAV-3′hNa_v_1.5-eGFP, which provided cardiac-specific overexpression of the dominant-negative form of hNa_v_1.5-R104W. At 8 weeks of age, echocardiography showed that mice with hNa_v_1.5-R104W had an increased left ventricular diameter (systolic and diastolic), end-diastolic volume, and stroke volume and a decreased left ventricular ejection fraction and fractional shortening compared with control mice. Overexpression of a dominant-negative form of hNa_v_1.5-R104W was responsible for the decreased heart rate and increased P wave duration on the electrocardiogram. Western blotting demonstrated reduced levels of total Nav1.5 protein in mice overexpressing hNa_v_1.5-R104W compared with controls. Patch-clamping of isolated adult cardiomyocytes revealed that hNa_v_1.5-R104W overexpression reduced Na^+^ current by 15% at −30 mV compared with control cardiomyocytes. Thus, it can be assumed that the mutant form of hNa_v_1.5-R104W causes decreased expression of the endogenous WT Na^+^-current channel and, as a consequence, there is a decrease in the Na^+^ current in the sarcoplasmic reticulum of cardiomyocytes—this fact may indicate the validity of the described animal model in the context of autosomal dominant Brugada syndrome in humans [233].

A model of acquired DCMP developing under oxidative stress in Wistar rats using a genetically encoded tool is described in detail. The plasmid cTnT-HyPer7-DAAO was used, providing cardiac-specific expression of a fusion protein complex—biosensor Hyper7 and yeast D-amino acid oxidase (DAAO). These genetic constructs as part of AAV9 were injected intravenously into 3–4-month-old male rats, followed by the addition of 1 M D-alanine to their drinking water for 4–6 weeks. D-alanine under the action of DAAO is oxidized to alpha-keto acid with the formation of the by-product H_2_O_2_. The Hyper7 biosensor, in turn, changes fluorescence characteristics in response to changes in the intracellular concentration of H_2_O_2_, enabling characterization of the extent of developing oxidative stress in the heart [234,235,236]. Rats with activated oxidative stress 4–6 weeks after transduction developed symptoms of DCMP compared with control animals: an increase in end-diastolic volume combined with a decrease in ejection fraction. Analysis of isolated papillary muscles with cTnT-HyPer7-DAAO showed reduced systolic strain in response to diastolic exercise, as well as a low physiological reaction to stimulation with beta-adrenergic receptor agonists. In the myocardium of DCMP rats, there was a decrease in the level of the phosphorylated form of PLB, decreased expression of the alpha isoform of myosin heavy chain (*MYH6*), and increased expression of brain and atrial natriuretic peptide (BNP and ANP) accompanied by increased activity of nuclear factor Nrf2. Animals with chronic DAAO activity in the heart showed increased plasma ANP and cardiac troponin I cTnI (cTnI) compared with controls. In the hearts of DAAO-expressing rats, a decrease in reduced glutathione (GSH) of 2 and increased total glutathione were detected compared with control animals. These changes may indicate a pathological level of oxidative stress development in model rats, which is characteristic of other cardiovascular diseases with the development of heart failure. Interestingly, the described model did not reproduce severe heart failure with the development of fibrotic remodeling of the ventricular configuration, which was confirmed by histological and molecular methods of investigation [235,236]. The rat model of heart failure following DCMP using the cTnT-HyPer7-DAAO plasmid was successfully used to study the therapeutic effect of the drugs sacubitril and valsartan [237].

Sometimes, genetically encoded tools are used to complement other existing methods of modeling cardiovascular disease, such as transgenic and surgical models (Figure 5). For example, the plasmid AAV9-ANF-Cre, which enables specific expression of Cre-recombinase in atrial cardiomyocytes through the atrial natriuretic hormone promoter, was used to activate *JPH2* (junctophilin-2; OMIM: 605267) gene knockdown via siRNA-mediated mechanism in transgenic mouse atria. After administration of the mentioned plasmids in mice, we observed disruption of normal sarcoplasmic Ca^2+^ turnover and more frequent Ca^2+^ sparking exclusively in mouse atria; the described changes in Ca^2+^ turnover did not affect ventricular cardiomyocytes [24].

Problems with delivery: Despite the above-mentioned advantages of animal models derived from genetically encoded constructs, this method is not yet widely used in biomedical and preclinical research. This fact can be explained by several reasons. Firstly, in experimental biomedicine and biotechnology, there is a substantial number of studies that can be performed on classical animal models, including phenotypically selected strains and transgenic animals. Consequently, genetically encoded tools for disease modeling are used only if classical methods do not provide the desired results. Secondly, genetically encoded tools are limited in their ability to be delivered into adult animals. Further in this section, the existing delivery challenges and their potential solutions will be discussed.

The most prevalent method for in vivo delivery of genetically encoded constructs involves packaging them into AAV (adeno-associated virus) vectors and subsequently injecting them into the systemic bloodstream. However, a challenge associated with viral delivery is the variability in tropism exhibited by viruses with distinct capsids for cardiomyocytes in the heart. The only AAV 1 vector that has undergone clinical trials has not demonstrated efficacy in patient treatment [238].

To date, AAV9 has been identified as the most effective viral serotype for vector delivery into rodent cardiomyocytes [239,240], compared to AAV1 and AAV6 serotypes. However, this type of virus accumulates in the liver of the model organism, demonstrating off-target effects [240]. The mitigation of these off-target effects can be achieved by selecting the correct cardiac-specific promoter, which will enable expression exclusively in the desired tissue type. Therefore, it is feasible to select promoters that will be active exclusively in atrial or ventricular cardiomyocytes, as the pattern of active genes in these cell types differs. Two promoters are shared by all cardiac cardiomyocyte types: cTnT and α-MHC [241]. The first one is the most prevalent for targeted expression in murine cardiomyocytes. The application of the cTnT promoter to the target genetic construct results in the expression of the product exclusively in the heart, as demonstrated by Prasad et al. [242]. This methodology enables the suppression of off-target transcription in other organs of the model animal.

Nonetheless, when AAV serotype 9 is utilized for gene delivery in conjunction with a cardiac-specific promoter to prevent off-target effects, a significant challenge persists. To achieve the targeted effect at the organ level, the genetic construct must enter and be expressed in every cell of the organ. Failure to achieve this effect leads to the development of a mosaic type of disease, wherein the organ consists of cells with the disease phenotype and wild-type cells. The issue of biodistribution of genetic constructs to different organs is a concern [243]. To increase the number of successfully transduced cells in the heart, it is possible to increase the number of viral particles delivered through the systemic bloodstream. However, this would result in both an increase in off-target effects in the animal body and an increase in the cost of such an animal model of the disease. Conversely, the development of synthetic viral serotypes exhibiting enhanced tropism for cardiomyocytes is a promising approach, as it is predicted to lead to more effective transfection of target cells. For instance, two novel viruses, AAV2-THGTPAD and AAV2- NLPGSGD, have been recently engineered based on AAV serotype 2, demonstrating superiority over both the paternal serotype and the most prevalent AAV 9 [244]. Thus, in comparison to AAV9, AAV2-THGTPAD and AAV2-NLPGSGD exhibit an approximately four-fold enhancement in tropism toward cardiomyocytes, as determined by the ratio of expression in the heart to expression in the liver.

Consequently, the creation of animal models of cardiovascular disease based on genetically encoded tools still faces certain challenges, which hinders their widespread use in biomedical research to some extent. Among the encountered issues, we would first of all mention off-target effects and the distribution of transduced cells throughout the whole organ. These problems are now being addressed by synthetic biology and the creation of viruses with increased tropism to tissue, as well as the use of tissue-specific promoters.

## 5. Conclusions and Perspectives

The examination of genetic cardiovascular models presented in this review highlights their essential role in advancing our understanding of cardiovascular pathophysiology and disease mechanisms. From spontaneous mutation models to precision genetic engineering approaches, these models have progressively enhanced our ability to investigate complex cardiovascular disorders with increasing translational relevance.

We would like to emphasize again that none of the current animal models of cardiovascular diseases can fully replicate the pathogenesis of these conditions in humans. Each model has its certain limitations. The main objective of our review was to provide researchers with some sort of a guide that could help them to select a genetic animal model that aligns with their research goals most closely.

The proposed algorithm for selecting genetic animal models for cardiovascular disease research: When selecting a specific model for studying any multifactorial disease, it is important to consider several aspects: physiological relevance, genetic manipulability, reproducibility, and translational potential (Figure 5).

Inbred models are most suitable for physiologically relevant modeling of multifactorial cardiovascular diseases (e.g., hypertension and non-hereditary forms of atherosclerosis), since the symptoms of the disease are determined by a combination of multiple mutant forms of various tissue-specific genes, which determines the completeness of the clinical picture of the pathology. For example, the development of arterial hypertension in SHR rats is caused by complex changes in the genetic apparatus of the cells of the heart, kidneys, blood vessels, and the vasomotor center of the medulla oblongata. Thus, such models have good translational potential for a more detailed and multifaceted study of the pathogenesis of vascular and heart diseases, as well as for the discovery of new target molecules and candidate genes for further possible therapeutic interventions. However, inbred models have a number of disadvantages—there are difficulties in terms of genetic manipulation and reproducibility, since close inbreeding can result in new undesirable mutations that are difficult for researchers to control and may cause uncontrollable pathological symptoms.

To enhance the possibility of genetic manipulation and controlled reproducibility of a specific factor under study in the pathogenesis of cardiovascular diseases, various transgenic models or models using genetically encoded tools in combination with various delivery systems are being actively developed. Compared to inbred models, when creating constitutive transgenic animals, researchers can accurately induce modifications of specific genes (their knockout, knockdown, overexpression, etc.) responsible for the development of a specific pathological feature of the disease in specific organs, tissues, and cells and also achieve reproducibility of these modifications across several generations of animals with great success. In addition, the use of, for example, genetically encoded tools with their delivery systems or the application of the Cre-LoxP system in models allows for fairly precise control of gene modifications in space and time in model animals. However, such models have limited physiological relevance and translational potential, since the modification of one or more genes does not allow the resulting transgenic model to be equated with many common multifactorial cardiovascular diseases, which are caused by changes in a whole set of genes in humans.

Evolution and contributions of genetic cardiovascular models: Classical genetic strains derived from spontaneous mutations have established foundational principles regarding genetic contributions to cardiovascular pathologies. Hypertensive rat models including SHR, DSS, FHH, and MHS have demonstrated how distinct genetic alterations can converge on similar phenotypes through diverse pathophysiological mechanisms: neurogenic pathways in SHR rats, salt sensitivity in DSS rats, and renal dysfunction in FHH and MHS rats. This pathophysiological heterogeneity mirrors the complex etiology of human cardiovascular diseases. Transgenic and knockout technologies have further refined cardiovascular modeling by enabling targeted investigation of individual genes and regulatory pathways implicated in cardiac pathologies. These approaches have proven valuable in elucidating the molecular mechanisms underlying monogenic disorders such as familial cardiomyopathies and channelopathies. The emergence of genetically encoded tools represents a transformative advancement in cardiovascular research. Viral vector systems, particularly adeno-associated virus (AAV) vectors, have enabled cardiac-specific gene delivery with remarkable efficiency. Similarly, CRISPR-Cas9 technology has revolutionized the field by enabling precise genome editing, facilitating the creation of models that more faithfully recapitulate human cardiovascular conditions.

Current challenges in genetic cardiovascular modeling: Despite significant advances, critical challenges persist in genetic modeling of cardiovascular pathology. The translational gap between animal models and human disease remains substantial, particularly for complex, multifactorial conditions. Species-specific differences in cardiovascular physiology necessitate cautious interpretation when extrapolating findings to human pathology. Moreover, the focus on single-gene modifications often fails to capture the polygenic architecture of most cardiovascular disorders. Methodological limitations also warrant consideration. Off-target effects in gene-editing approaches, variability in transgene expression levels, and incomplete penetrance of phenotypes can confound experimental results. Another significant challenge involves accurately modeling the temporal aspects of cardiovascular disease progression, as human pathologies typically develop over the influence of multiple environmental factors and comorbidities.

Future directions and technological innovations: The future of genetic modeling of cardiovascular diseases lies in integrative approaches and technological innovation. Multi-omics analyses combined with genetic models will enable comprehensive mapping of disease networks, advancing toward systems-level understanding. The integration of transcriptomics, proteomics, and metabolomics with precise genetic modifications will provide unprecedented insights into the molecular signatures of cardiovascular pathologies. The development of humanized cardiovascular models represents another promising frontier. By incorporating human genes, cells, or tissues into animal models, researchers can bridge critical species gaps and create more translation experimental systems. Advances in genome editing precision will fundamentally enhance model fidelity. Next-generation CRISPR systems with reduced off-target effects, base editing capabilities, and prime editing technologies will enable increasingly sophisticated genetic modifications. The integration of genetic modeling with complementary technologies such as optogenetics and genetically encoded biosensors presents extraordinary opportunities for cardiovascular research by enabling dynamic monitoring of cellular processes in living animals.

Translational impact and clinical applications: The ultimate objective of genetic cardiovascular modeling is to advance translational medicine and improve human health outcomes. Refined genetic models will facilitate more accurate identification of therapeutic targets, enable robust preclinical evaluation of drug efficacy, and support the development of precision medicine approaches for cardiovascular diseases. Particularly promising is the emerging interface between genetic animal models and human stem cell biology. Patient-specific induced pluripotent stem cells (iPSCs) harboring disease-causing mutations, coupled with directed differentiation into cardiovascular lineages, offer complementary approaches to traditional animal models. The application of genetic models in drug development has already yielded significant advances in cardiovascular therapeutics. From PCSK9 inhibitors for hypercholesterolemia to novel anti-arrhythmic approaches targeting specific ion channels, insights from genetic models have directly translated into clinical interventions.

Conclusion: Genetic animal models have been instrumental in deciphering the complex pathophysiology of cardiovascular diseases and will continue to evolve alongside technological advances. While acknowledging their inherent limitations, these models remain essential tools for basic and translational cardiovascular research. The future lies in creating increasingly sophisticated models that better reflect human cardiovascular pathology, thereby enhancing their translational value. The integration of genetic models with complementary approaches, including computational modeling and human tissue systems, will be critical for comprehensive understanding and effective intervention in cardiovascular pathologies. The continued refinement of genetic cardiovascular models, coupled with advances in phenotyping technologies, promises to accelerate discovery in cardiovascular medicine and ultimately improve outcomes for patients.

## Figures and Tables

**Figure 1 biomedicines-13-01518-f001:**
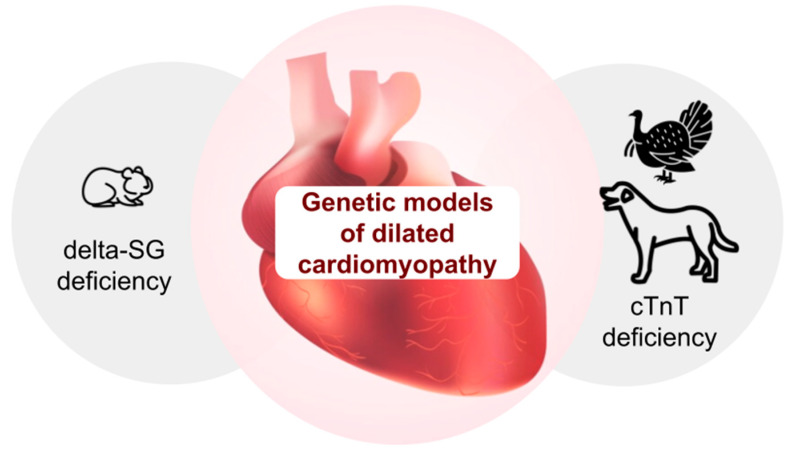
Genetic models of dilated cardiomyopathy (DCMP) include Syrian golden hamster model of deficiency of delta-sarcoglycan (delta-SG) and domestic turkey and dog models of deficiency of cardiac troponin T (cTnT).

**Figure 2 biomedicines-13-01518-f002:**
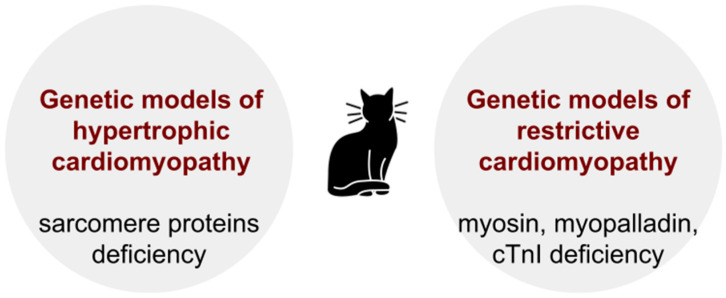
Genetic models of hypertrophic cardiomyopathy (HCMP) and restrictive cardiomyopathy (RCMP). HCMP cat models (Maine Coon cat, Persian, Ragdoll, and Sphynx) of deficiency of sarcomere proteins encoded by *MYOC*, *MYBPC* genes. RCMP cat models of deficiency of myosin, myopalladin, and cardiac troponin I (cTnI).

**Figure 3 biomedicines-13-01518-f003:**
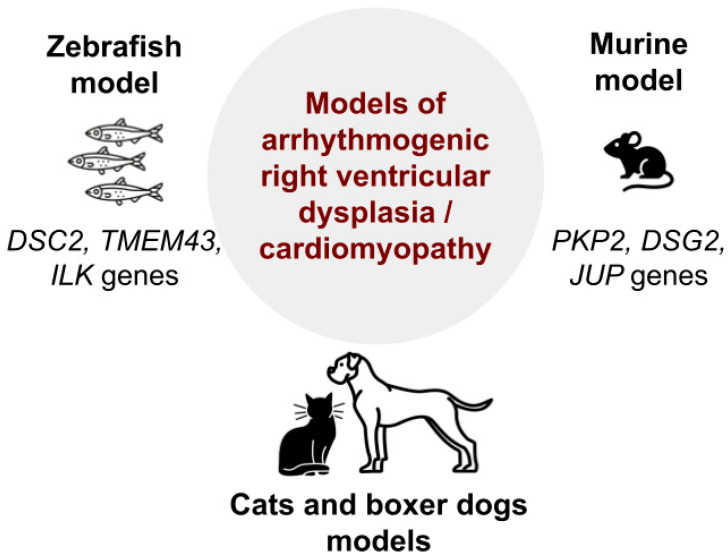
Animal models of arrhythmogenic right ventricular dysplasia/cardiomyopathy (ARVC) include murine models with abnormalities in genes *PKP2*, *DSG2*, and *JUP*; zebrafish models with abnormalities in genes *DSC2*, *TMEM43*, and *ILK*; and spontaneous ARVC cat and boxer dog models.

**Figure 4 biomedicines-13-01518-f004:**
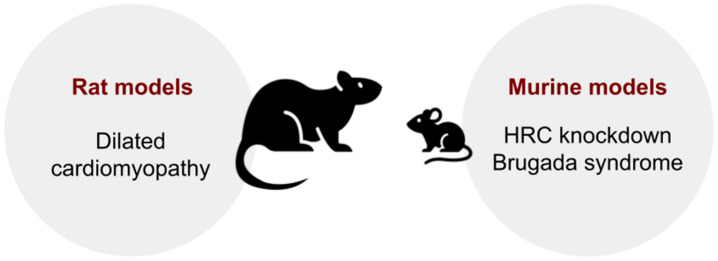
Specific examples of animal models of cardiovascular disease using genetically encoded tools.

**Figure 5 biomedicines-13-01518-f005:**
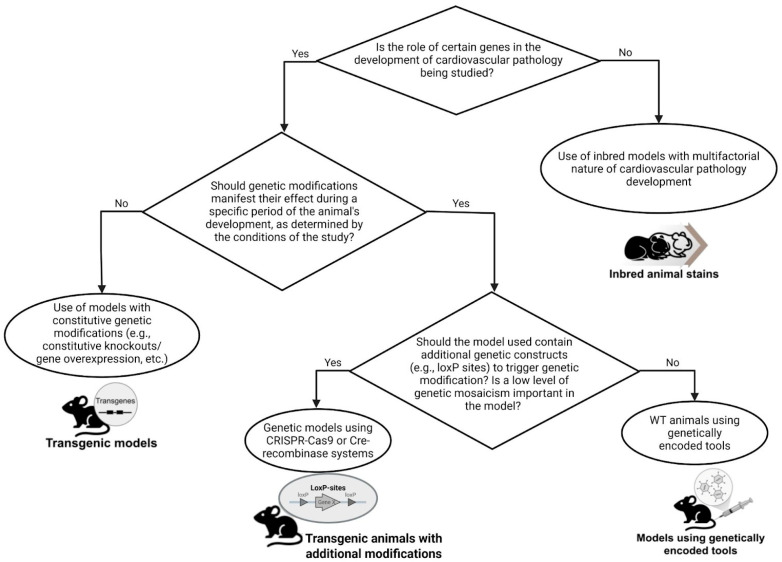
Proposed algorithm for choosing genetic animal models for studying cardiovascular diseases.

**Table 1 biomedicines-13-01518-t001:** Comprehensive summary of genetic animal models for cardiovascular diseases.

Model	Genetic Basis/Modification	Main Cardiovascular Phenotype	Key Applications	Major Advantages	Main Limitations	Physiological Relevance	**Genetic Manipulability**	**Reproducibility**	**Translational Relevance**
SHR Rat	Polygenic (overexpression of the renin gene)	Essential hypertension, stroke-prone	Essential hypertension, cerebral circulatory disorders	Closely mimics human essential hypertension	Limited to neurogenic hypertension	High	Moderate	High	Moderate-High
SHR-SP Rat	Polygenic (derived from SHR)	Severe hypertension, stroke susceptibility	Stroke research	Reproducible stroke phenotype	Limited lifespan	High	Moderate	High	High
DSS Rat	Polygenic *(CYP11B1)*	Salt-sensitive hypertension	Sodium-related hypertension	Gene-environment interaction modeling	No human-like nephropathy	Moderate	Low	High	Moderate
FHH Rat	Polygenic *(Add3, Rbm20, Shoc2)*	Hypertensive nephropathy	Renal hypertension	Renal-cardiovascular crosstalk	Unclear primary pathology	Moderate	Low	Moderate	Moderate
MHS Rat	*ADD1* mutation (α-adducin defect)	Primary hypertension with renal dysfunction	Hypertension with kidney involvement	Links hypertension to kidney pathology	Limited non-renal applications	Moderate	Low	High	Moderate
Milan Hypertensive Rat	Polygenic (α-adducin mutations)	Essential hypertension	Membrane transport studies	Well-characterized genetic basis	Limited availability	Moderate	Low	High	Moderate
Lyon Hypertensive Rat	Polygenic *(Ercc6l2)*	Metabolic syndrome, salt-sensitive hypertension	Insulin resistance studies	Human metabolic syndrome mimic	Complex genetic architecture	Moderate	Low	Moderate	Low
NZGH Rat	Polygenic selection	Spontaneous hypertension	Hypertension mechanisms	Stable phenotype	Low genetic tractability	Moderate	Low	High	Moderate
MWF Rat	Spontaneous renal dysfunction	Hypertension with proteinuria	Renal hypertension	Kidney-heart axis modeling	Slow progression	Moderate	Low	Moderate	Low
Sabra Hypertensive Rat	α2-Adrenoceptor variants	Salt-sensitive hypertension	Environmental hypertension	Human salt sensitivity reproduction	Limited genetic tools	Moderate	Low	High	Moderate
Buffalo Rat	Polygenic	Hypertension with insulin resistance	Metabolic hypertension	Metabolic-cardiovascular interactions	Complex phenotype	Moderate	Low	Moderate	Moderate
Goto-Kakizaki Rat	Polygenic	Diabetic cardiomyopathy	Diabetes-related CVD	Non-obese diabetes model	Mild cardiac phenotype	Moderate	Low	High	Moderate
WHHL Rabbit	*LDLR* mutation	Familial hypercholesterolemia	Atherosclerosis studies	Spontaneous human-like plaques	High cost, limited tools	High	Moderate	High	High
Ldlr^−/−^ Mouse	*LDLR* knockout	Hypercholesterolemia	Plaque biology research	Rapid disease progression	Species-specific plaque differences	Moderate	High	High	Moderate
ApoE^−/−^ Mouse	ApoE knockout	Severe atherosclerosis	Plaque development studies	Widely used model	Plaque composition differences	Moderate	High	High	Moderate
cMyBP-C KO Mouse	*MYBPC3* deletion	Hypertrophic cardiomyopathy	Sarcomere dysfunction	Human mutation similarity	Mouse-specific physiology	Moderate	High	High	Moderate
Feline HCM Model	*MYBPC3* mutation (natural)	Hypertrophic cardiomyopathy	Spontaneous HCM studies	Naturally occurring HCM	Limited genetic manipulation	High	Low	Moderate	High
Canine ARVC Model	*PKP2* mutation (boxers)	Arrhythmogenic right ventricular cardiomyopathy	ARVC mechanisms, sudden cardiac death	Naturally mimics human ARVC	Ethical and cost challenges	High	Low	Moderate	High
Zebrafish (Tnnt2 KO)	*Tnnt2* knockout	Cardiomyopathy	Cardiac development	High-throughput screening	Simplified cardiac structure	Low-Moderate	High	High	Low
CPVT Mouse	*RYR2* mutation	Polymorphic VT	Arrhythmia mechanisms	Human CPVT recapitulation	Stress-dependent phenotype	Moderate	High	High	Moderate
LQTS Mouse	*KCNQ1/KCNH2* KO	Long QT syndrome	Ion channel studies	Direct human mutation link	ECG differences	Moderate	High	High	Moderate
AAV9-MYH7 Mouse	AAV9-mediated *MYH7* mutation	Hypertrophic cardiomyopathy	Gene therapy testing	Tissue-specific targeting	Transient expression	Moderate	High	Moderate	High
DREADD Rat	Chemogenetic hM3Dq receptor expression	Heart failure modulation	Neural regulation of cardiac function	Precise control of signaling pathways	Requires ligand administration	Moderate	High	High	Moderate

**Table 2 biomedicines-13-01518-t002:** The comparative advantages and disadvantages of popular animal species used for modeling heart diseases.

Animal Model	Advantages	Drawbacks
Danio rerio (zebrafish)	Short developmental period, low-cost maintenance	Two-chamber heart, which do not reproduce all processes typical for humans
Mus musculus (house mouse)	Short developmental period, low-cost maintenance	Small-size heart, high heartbeat rate
Rabbit	Short developmental period, high efficacy of gene modifications	Moderate maintenance costs

## Data Availability

Not applicable.

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
