# Peer review of "Genetic Animal Models of Cardiovascular Pathologies"

_biomedicines, 2025, doi:10.3390/biomedicines13071518_

Round 1
Reviewer 1 Report
Comments and Suggestions for Authors
The review article 'Genetic Animal Models of Cardiovascular Pathologies' summarize the major animal models for cardiomyopathies, Hypertension, CAD and Channelopathies.
The topic of this review article is interesting to a broad readership. However, I suggest several points which should be changed before this manuscript should be published:
1.) I would add for each human gene an OMIM identifier.
2.) Human gene names are not always written in Italics according to the official guidelines.
3.) Line 278 and 281: MYBPC3 (the 3 is missing).
4.)Line 314-317: Please add also the TMEM43 and ILK genes to this list (see 'Arrhythmogenic right ventricular cardiomyopathy type 5 is a fully penetrant, lethal arrhythmic disorder caused by a missense mutation in the TMEM43 gene' and
'Mutations in ILK, encoding integrin-linked kinase, are associated with arrhythmogenic cardiomyopathy' 2019
5.) I would add also the zebrafish models for TMEM43 and ILK in Figure 4 and would also reference and explain them in the text.
See ‘Altered Expression of TMEM43 Causes Abnormal Cardiac Structure and Function in Zebrafish’ Zink 2022.
In summary, I think that a major revision of this manuscript is necessary. However, I am optimistic that the authors can fix the criticized points.
Author Response
Comments and Suggestions for Authors
The review article 'Genetic Animal Models of Cardiovascular Pathologies' summarize the major animal models for cardiomyopathies, Hypertension, CAD and Channelopathies. The topic of this review article is interesting to a broad readership. However, I suggest several points which should be changed before this manuscript should be published.
Dear Reviewer! Thank you so much for the careful analysis of our paper and useful remarks which, we hope, enabled us to improve it. We tried to give the most accurate answers for each of the points of your comments. All changes are highlighted in the text in color.
Comments 1: I would add for each human gene an OMIM identifier.
Response 1: We have added OMIM identifiers for all human genes mentioned.
Comments 2: Human gene names are not always written in Italics according to the official guidelines.
Response 2: We have corrected this default.
Comments 3: Line 278 and 281: MYBPC3 (the 3 is missing).
Response 3: We have corrected this mistake.
Comments 4: Line 314-317: Please add also the TMEM43 and ILK genes to this list (see 'Arrhythmogenic right ventricular cardiomyopathy type 5 is a fully penetrant, lethal arrhythmic disorder caused by a missense mutation in the TMEM43 gene' and 'Mutations in ILK, encoding integrin-linked kinase, are associated with arrhythmogenic cardiomyopathy' 2019
Response 4: We have added these links to the text.
Comments 5: I would add also the zebrafish models for TMEM43 and ILK in Figure 4 and would also reference and explain them in the text. See ‘Altered Expression of TMEM43 Causes Abnormal Cardiac Structure and Function in Zebrafish’ Zink 2022.
Response 5: We have updated Figure 4 adding TMEM43 and ILK genes to Zebrafish models. We have also made corresponding explanation with references in the text
In summary, I think that a major revision of this manuscript is necessary. However, I am optimistic that the authors can fix the criticized points.
Reviewer 2 Report
Comments and Suggestions for Authors
Mikhail Blagonravov et al. provides a comprehensive overview of genetic animal models utilized in cardiovascular disease research. The manuscript encompasses a wide range of models, spanning from spontaneously hypertensive strains to cutting-edge genome editing techniques. Its strength lies in its systematic integration of the advantages and limitations of each model, with a specific emphasis on the applications and potential of emerging technologies such as AAV-mediated gene delivery and CRISPR-Cas9. The authors' focus on the translational relevance of these models further enhances the manuscript's overall impact and utility for researchers in the field. While the review offers a valuable contribution, there are several key areas that require further attention:
1. The methods section lacks sufficient detail regarding the literature search and selection process. Clear articulation of the databases searched, keywords used, inclusion/exclusion criteria, and methods for data extraction is essential to ensure reproducibility and minimize potential bias.
2. The criteria used for evaluating the various animal models and gene editing technologies are not explicitly defined. A more rigorous and standardized framework for assessing models based on factors such as physiological relevance, genetic manipulability, reproducibility, and translational potential is needed. This framework should then be applied consistently throughout the review.
3. The discussion section could benefit from a more in-depth exploration of the limitations of existing models and how emerging technologies can address these shortcomings. Furthermore, a more critical analysis of the challenges associated with translating findings from animal models to human diseases is warranted.
4. The manuscript would be enhanced by the inclusion of more tables and figures to visually represent key information. For example, a comprehensive table summarizing the characteristics, applications, and limitations of different animal models would be particularly useful.
Minor comments:
By addressing these concerns and incorporating the suggested revisions, the authors can significantly enhance the quality and impact of their review article, making it a valuable resource for the cardiovascular research community.
Line 53, "each of these model groups has its advantages and limitations" should be "each of these model groups have its advantages and limitations"
Line 56, "can fully reproduce the pathogenesis of the corresponding human pathology", "fully reproduce' to "entirely replicate" to reduce the potential misunderstanding of absolute wording.
Line 73, "investigations in the field of cardiac pathophysiology in small mammals" should be "small mammal studies in the field of cardiac pathophysiology"
Line 104-105, "inbred strains that known to reproduce" should be "inbred strains that are known to reproduce"
Line 133-134, "a strain of rats’ sensitive to high-salt diet was selected." should be "a strain of rats sensitive to high-saltdiet was selected."
Line 157, "hypertension in this rat strain is linked" should be "The hypertension in this rat strain is linked..."
Line 164-178, "the Lion hypertensive rats" and "the Lion hypertensive rats" are the same rats?
Line 179, "the Sabra hypertensive rats is" should be "the Sabra hypertensive rats are"
Line 199-200, "Munich Wistar Frömter (MWF) rats, which is a model"should be "Munich Wistar Frömter (MWF) rats, which are a model."
Line 247, "In domestic turkeys, in 2-5% of cases, DCMP develops within the first 4 weeks of life"occurs in the absence of other cardiac abnormalities. This condition is characterized by dilated cardiomyopathy, leading to heart failure and high mortality rates in affected turkeys. The underlying causes are not fully understood but may involve genetic predisposition and environmental factors.
Line 305, "since, as mentioned previously (Fig. 4), the pathogenic, morphological, and clinical patterns of these diseases more closely resemble human cardiomyopathy compared with transgenic models." Split or adjust the sentence into two sentences to make the logic clearer, as "Since, as previously mentioned (Fig. 4), the pathogenic, morphological, and clinical patterns of these diseases more closely resemble human cardiomyopathy than transgenic models, it is a more suitable approach for translational studies."
Line 333-334, "in 8 animals they diagnosed right ventricular congestive heart failure with supraventricular, ventricular tachyarrhythmia." shoule be "in 8 animals, they diagnosed right ventricular congestive heart failure with supraventricular and ventricular tachyarrhythmias."
Line 388-389, "Therefore, when creating transgenic animal strains using the random integration method, multiple strains with the same transgene insertion are typically generated, and a strain with the desired level of transgene expression is subsequently selected." The sentence is too long and split into two sentences for clarity. This would ensure better readability and comprehension, as well as prevent any confusion that may arise from overly complex sentence structures.
Line 627, "..... the enhance of survival rate...." The phrase "the enhance" has been changed to "the enhancement".
Author Response
Comments and Suggestions for Authors
Mikhail Blagonravov et al. provides a comprehensive overview of genetic animal models utilized in cardiovascular disease research. The manuscript encompasses a wide range of models, spanning from spontaneously hypertensive strains to cutting-edge genome editing techniques. Its strength lies in its systematic integration of the advantages and limitations of each model, with a specific emphasis on the applications and potential of emerging technologies such as AAV-mediated gene delivery and CRISPR-Cas9. The authors' focus on the translational relevance of these models further enhances the manuscript's overall impact and utility for researchers in the field. While the review offers a valuable contribution, there are several key areas that require further attention:
Dear Reviewer! Thank you so much for your valuable remarks! We have tried to do our best to revise the article in accordance with them. All changes are highlighted in the text in color.
Comments 1: The methods section lacks sufficient detail regarding the literature search and selection process. Clear articulation of the databases searched, keywords used, inclusion/exclusion criteria, and methods for data extraction is essential to ensure reproducibility and minimize potential bias.
Response 1: We completely agree that the methodological section should be expanded and supplemented. We have done it according to the Reviewer’ recommendations and added the corresponding fragment to Introduction:
The process of preparing this review involved employing several methodical approaches to the search and selection of literature sources. The search was performed using databases, including, first of all, PubMed, Google Scholar and Research Gate. When searching for publications, the following keywords were used (both separately and in combination) in general: genetic models; animal models, mice, murine, rat, rabbit, cat, feline, dog, zebrafish, transgenic; knockout; hypertension; cardiomyopathy; atherosclerosis; arrhythmias; adeno-associated dependoparvovirus A; CRISPR associated protein 9. Depending on the particular section, we employed the following terms for our search: Chapter 2 – genetic models, animal models, inbred strains, heart, cardiac, hypertension, cardiomyopathy, atherosclerosis, coronary heart disease; Chapter 3 – genetic models, animal models, transgenic, knockout, heart, cardiac, CRISPR/Cas9; cardiac arrhythmias, channelopathies, cardiomyopathy, atherosclerosis, myocardial hypertrophy; Chapter 4 – genetic models, heart, cardiac, genetically encoded tolls, adeno-associated dependoparvovirus A, transduction, knockout, Cre-LoxP system, plasmid. Priority was given to the publications over the last 5-10 years. In total, more than 900 sources were analyzed and 244 of them were included in the final list of references for the review. The inclusion criteria were as follows: publications strictly related to the topic of the review (genetic models of cardiovascular disease), sufficient breadth and depth of coverage, relevance and reliability of sources (only articles from scientific journals and monographs were considered), as well as accessibility to the full-text version. And we also applied the following exclusion criteria: irrelevant sources, publications with methodological flaws, and sources that did not correspond to the methodology we were analyzing (we were interested in experimental research-based publications).
Comments 2: The criteria used for evaluating the various animal models and gene editing technologies are not explicitly defined. A more rigorous and standardized framework for assessing models based on factors such as physiological relevance, genetic manipulability, reproducibility, and translational potential is needed. This framework should then be applied consistently throughout the review.
Response 2: We are very grateful to the reviewer for raising an important issue of the lack of clear criteria for evaluating animal models and gene editing technologies. We acknowledge this limitation in our current review structure and fully agree that that establishing standardized evaluation criteria based on physiological relevance, genetic manipulability, reproducibility of experiments, and translational potential is essential for providing meaningful recommendations to the cardiovascular research community. To address this issue, we have implemented the recommended multidimensional evaluation system and presented it in a comprehensive summary table that allows direct comparison of animal models across multiple criteria, facilitating evidence-based model selection for specific research objectives (Table 1 has been significantly expanded).
Comments 3: The discussion section could benefit from a more in-depth exploration of the limitations of existing models and how emerging technologies can address these shortcomings. Furthermore, a more critical analysis of the challenges associated with translating findings from animal models to human diseases is warranted.
Response 3: We agree that the limitations of existing animal models and the analysis of the possibilities of overcoming them using modern technological solutions is an important aspect of our review. In accordance with the Reviewer's recommendations reflected in Comments 2 and 4, we considered it possible to present them in an expanded version of the table. The challenges associated with transferring the results obtained in animal models to human diseases are described in section 5. ‘Conclusions and perspectives’, in particular, in the subsections ‘Current challenges in genetic cardiovascular modeling’, ‘Future directions and technological innovations’ and ‘Translational impact and clinical applications’. At the same time, since the main objective of our review was to provide researchers with some sort of a guide that could help them to select a genetic animal model that aligns with their research goals most closely, we decided that it would be appropriate to supplement this section with a diagram reflecting the algorithm for searching for animal models in the frame of discussion. We have made a corresponding additional figure (Fig. 6) and accompanied it with a detailed comment in the text of this section. All changes and additions have been incorporated in the revised version of the manuscript.
Comments 4: The manuscript would be enhanced by the inclusion of more tables and figures to visually represent key information. For example, a comprehensive table summarizing the characteristics, applications, and limitations of different animal models would be particularly useful.
Response 4: We would like to thank the Reviewer for the valuable suggestion to supplement the manuscript with additional tables and figures, in particular a comprehensive table summarizing the characteristics, areas of application, and limitations of various animal models. We agree that visual representation of key information will greatly enhance the clarity, accessibility, and practical value of our review. We have added a detailed summary table of animal models to the manuscript and presented a model selection algorithm (decision tree flowchart) that provides a step-by-step visual guide for researchers.
Minor comments:
By addressing these concerns and incorporating the suggested revisions, the authors can significantly enhance the quality and impact of their review article, making it a valuable resource for the cardiovascular research community.
Comments 1: Line 53, "each of these model groups has its advantages and limitations" should be "each of these model groups have its advantages and limitations"
Response 1: Completed.
Comments 2: Line 56, "can fully reproduce the pathogenesis of the corresponding human pathology", "fully reproduce' to "entirely replicate" to reduce the potential misunderstanding of absolute wording.
Response 2: Done.
Comments 3: Line 73, "investigations in the field of cardiac pathophysiology in small mammals" should be "small mammal studies in the field of cardiac pathophysiology"
Response 3: Completed.
Comment 4: Line 104-105, "inbred strains that known to reproduce" should be "inbred strains that are known to reproduce"
Response 4: Completed.
Comments 5: Line 133-134, "a strain of rats’ sensitive to high-salt diet was selected." should be "a strain of rats sensitive to high-salt diet was selected."
Response 5: Completed.
Comments 6: Line 157, "hypertension in this rat strain is linked" should be "The hypertension in this rat strain is linked..."
Response 6: Completed.
Comments 7: Line 164-178, "the Lion hypertensive rats" and "the Lion hypertensive rats" are the same rats?
Response 7: Yes, they are.
Comments 8: Line 179, "the Sabra hypertensive rats is" should be "the Sabra hypertensive rats are"
Response 8: Corrected.
Comments 9: Line 199-200, "Munich Wistar Frömter (MWF) rats, which is a model" should be "Munich Wistar Frömter (MWF) rats, which are a model."
Response 9: Completed.
Comments 10: Line 247, "In domestic turkeys, in 2-5% of cases, DCMP develops within the first 4 weeks of life" occurs in the absence of other cardiac abnormalities. This condition is characterized by dilated cardiomyopathy, leading to heart failure and high mortality rates in affected turkeys. The underlying causes are not fully understood but may involve genetic predisposition and environmental factors.
Response 10: We have added this extension.
Comments 11: Line 305, "since, as mentioned previously (Fig. 4), the pathogenic, morphological, and clinical patterns of these diseases more closely resemble human cardiomyopathy compared with transgenic models." Split or adjust the sentence into two sentences to make the logic clearer, as "Since, as previously mentioned (Fig. 4), the pathogenic, morphological, and clinical patterns of these diseases more closely resemble human cardiomyopathy than transgenic models, it is a more suitable approach for translational studies."
Response 11: We have changed this sentence according to the recommendation.
Comments 12: Line 333-334, "in 8 animals they diagnosed right ventricular congestive heart failure with supraventricular, ventricular tachyarrhythmia." should be "in 8 animals, they diagnosed right ventricular congestive heart failure with supraventricular and ventricular tachyarrhythmias."
Response 12: We have corrected this sentence.
Comments 13: Line 388-389, "Therefore, when creating transgenic animal strains using the random integration method, multiple strains with the same transgene insertion are typically generated, and a strain with the desired level of transgene expression is subsequently selected." The sentence is too long and split into two sentences for clarity. This would ensure better readability and comprehension, as well as prevent any confusion that may arise from overly complex sentence structures.
Response 13: We have divided this sentence into 2 ones as follows: “Therefore, when creating transgenic animal strains using the random integration method, multiple strains with the same transgene insertion are typically generated. In this case a strain with the desired level of transgene expression is subsequently selected.”
Comments 14: Line 627, "..... the enhance of survival rate...." The phrase "the enhance" has been changed to "the enhancement".
Response 14: We have corrected this mistake.
Round 2
Reviewer 1 Report
Comments and Suggestions for Authors
The authors have improved their manuscript in a convinving way. All my criticized points were changed and improved. Therefore I suggest to accept this manuscript for publication.